# Can Students Outperform Teachers in Knowledge Distillation based Model Compression?

## Abstract

Knowledge distillation (KD) is an effective technique to compress a large model (teacher) to a compact one (student) by knowledge transfer. The ideal case is that the teacher is compressed to the small student without any performance dropping. However, even for the state-of-the-art (SOTA) distillation approaches, there is still an obvious performance gap between the student and the teacher. The existing literature usually attributes this to model capacity differences between them. However, model capacity differences are unavoidable in model compression. In this work, we systematically study this question. By designing exploratory experiments, we find that model capacity differences are not necessarily the root reason, and the distillation data matters when the student capacity is greater than a threshold. In light of this, we propose to go beyond in-distribution distillation and accordingly develop KD+. KD+ is superior to the original KD as it outperforms KD and the other SOTA approaches substantially and is more compatible with the existing approaches to further improve their performances significantly [1].

## 1 Introduction

Deep neural networks (DNNs) have achieved remarkable performances in various domains, but they require large amounts of computation and memory. This seriously limits their deployment with limited resources or a strict latency requirement. One solution to this problem is knowledge distillation which transfers the knowledge from a large network (teacher) to a small one (student).

Hinton et al. (2015) proposed the original knowledge distillation [2] (KD) which uses softened logits of a teacher as supervision to train a student. To make the student better capture the knowledge from the teacher, the existing studies focus on aligning their representations by using different criteria. However, there is still a significant performance gap between the teacher and the student. Figuring out the reason for this gap is essential for further improving the student performance.

Mirzadeh et al. (2020) argue that the model capacity difference causes the failure for transferring the knowledge from a large teacher to a small student, thus leading to a large performance gap. Similarly, Cho & Hariharan (2019) point out that as the teacher grows in capacity and accuracy, it is difficult for the student to emulate the teacher. In this paper, we systematically study why students underperform teachers and how students can match or outperform teachers. We find that in most experimental settings of the existing literature, the root reason for the performance gap is not necessarily the capacity differenc as the student is powerful enough to memorize the teacher's outputs. The reason lies in the *distillation dataset* on which the knowledge is transferred.

As an old proverb says, indigo comes from blue, but it is bluer than blue. In reality, it is not rare for human students to do better than their teachers. These excellent human students not only well capture the knowledge from their teachers but also learn more related knowledge on their own. This gives an insight for students in KD to match or outperform their teachers. We find that currently the students in KD have not well captured the knowledge in their teachers as they only mimic the behavior of the teachers on sparse training data points. In light of this, we propose KD+ which goes beyond in-distribution distillation to substantially reduce the performance gap between students and teachers.

Our main contributions are summarized as follows:

---

[1] The code will be released online.

[2] In this paper, we use KD to denote the original knowledge distillation algorithm Hinton et al. (2015).

- Different from the common belief that model capacity differences result in the performance gap between students and teachers, we find that capacity differences are not necessarily the root reason and instead the distillation data matters when students' capacities are greater than a threshold. To our best knowledge, this is the first work that **systematically** explores why small students underperform teachers and how students can outperform large teachers.

- By designing exploratory experiments, we find the following: (1) only fitting teachers' outputs at sparse training data points cannot make students well capture the local, in-distribution shapes of the teacher functions; (2) different from the case on standard supervised learning, out-of-distribution data (but not all) can be beneficial to knowledge distillation.

- Different from the existing work focusing on using different criteria to align representations or logits between teachers and students, we address knowledge distillation from a novel (data) perspective by going beyond in-distribution distillation and accordingly develop KD+.

- Extensive experiments demonstrate that KD+ largely reduces the performance gap between students and teachers, and even enables students to match or outperform their teachers. KD+ is superior to KD as it outperforms KD and more than 10 SOTA methods substantially and shows a better compatibility with the existing methods and superiority in few-shot scenario.

## 2 RELATED WORK

The objective function of knowledge distillation can be simply expressed as a combination of the regular cross-entropy objective and a distillation objective. According to the distillation objective, the existing literature can be divided into logit-based approaches (Hinton et al., 2015) and representation-based approaches (Romero et al., 2015). Logit-based approaches construct the distillation objective based on output logits. Hinton et al. (2015) propose KD which penalizes the softened logit differences between a teacher and a student. Park et al. (2019) propose to transfer data sample relations from a teacher to a student by aligning their logit-based structures. On the other hand, representation-based approaches design the distillation objective based on feature maps. FitNet (Romero et al., 2015) aligns the features of a teacher and a student through regressions. AT (Zagoruyko & Komodakis, 2017) distills feature attention from a teacher into a student. CRD (Tian et al., 2020) maximizes the mutual information between student and teacher representations. Other representation-based methods (Yim et al., 2017; Huang & Wang, 2017; Kim et al., 2018; Liu et al., 2019; Srinivas & Fleuret, 2018; Wang et al., 2018; Heo et al., 2019a; Cho & Hariharan, 2019; Ahn et al., 2019; Koratana et al., 2019; Aguilar et al., 2019; Shen & Savvides, 2020) use different criteria to align feature representations. SSKD (Xu et al., 2020) introduces extra self-supervision tasks to assist KD. Online knowledge distillation (Zhang et al., 2018b; Chen et al., 2020; Anil et al., 2018; Chung et al., 2020; Zhu et al., 2018) trains multiple students simultaneously. Self-distillation (Furlanello et al., 2018; Yuan et al., 2020) approaches train a DNN by using itself as the teacher. It is observed that the existing studies focus on designing different criteria to align teacher-student representations or logits on in-distribution data. In this work, we address knowledge distillation from a data perspective by embedding out-of-distribution distillation into a regularizer.

Mirzadeh et al. (2020) observe that the model capacity gap results in the failure for transferring knowledge from a large teacher to a small student, thus causing a performance gap. To reduce this gap, they propose a multi-step knowledge distillation framework by using several intermediate-size networks (teacher assistants). However, the students still underperform the teachers substantially. Cho & Hariharan (2019) argue that as the teacher grows in capacity and accuracy, it is difficult for the student to emulate the teacher. To reduce the influence of the large capacity gap, they regularize both the teacher and the knowledge distillation by early stopping. We find that capacity differences are not necessarily the root reason when student capacities are greater than a threshold.

On the other hand, KD+ goes beyond in-distribution distillation by exploring the knowledge between two training samples. Similar techniques have been used in many applications with different goals and mechanisms. Mixup (Zhang et al., 2018a) enforces local linearity of a DNN by linearly interpolating a random pair of training samples and their one-hot labels simultaneously. However, simply interpolating two labels may not match the generated sample as pointed out in (Guo et al., 2019). KD+ does not have the above issue as it teaches a student to mimic the local shape of a powerful teacher. MixMatch (Berthelot et al., 2019b) linearly interpolates labeled and unlabeled data to improve the semi-supervised learning performances. ReMixMatch (Berthelot et al., 2019a) improves MixMatch

by introducing distribution alignment and augmentation anchoring. DivideMix (Li et al., 2020) aims to learn with noisy labels by modifying MixMatch with label co-refinement and label co-guessing on labeled and unlabeled samples, respectively. AugMix (Hendrycks et al., 2019) linearly interpolates original training samples and augmented training samples to improves the robustness and uncertainty estimates of DNNs.

## 3 REFORMULATING KD

Hinton et al. (2015) propose KD which minimizes the softened logit differences between a student and a teacher over training data $D_t = (X_t, Y_t)$ where $X_t$ and $Y_t$ are the training samples and the ground truth, respectively. The complete objective is:

$$\mathcal{L}_{KD} = \sum_{(x_t, y_t) \in (X_t, Y_t)} [\alpha \mathcal{L}_{CE}(f_S, x_t, y_t) + \beta \mathcal{L}_{KL}(f_S, f_T, x_t)] \tag{1}$$

where $\alpha$ and $\beta$ are balancing weights and $\mathcal{L}_{CE}$ is the regular cross-entropy objective:

$$\mathcal{L}_{CE}(f_S, x_t, y_t) = H(y_t, \sigma(f_S(x_t))) \tag{2}$$

where $H(.)$ is the cross-entropy and $\sigma$ is softmax. $\mathcal{L}_{KL}$ in (1) is the distillation objective:

$$\mathcal{L}_{KL}(f_S, f_T, x_t) = \tau^2 KL \left( \sigma \left( \frac{f_T(x_t)}{\tau} \right), \sigma \left( \frac{f_S(x_t)}{\tau} \right) \right) \tag{3}$$

where $\tau$ is a temperature to generate soft labels and $KL$ represents KL-divergence. KD can be considered as using one function ($f_S$) to fit the outputs of another function ($f_T$).

We notice that in (1), $\mathcal{L}_{CE}$ requires both data samples $X_t$ and the corresponding ground truth $Y_t$ while $\mathcal{L}_{KL}$ only needs data samples $X_t$ for distilling the teacher knowledge. In light of the difference, **we consider KD from semi-supervised perspective** and reformulate (1) in a more general form:

$$\mathcal{L} = \sum_{(x_t, y_t) \in (X_t, Y_t)} \alpha \mathcal{L}_{CE}(f_S, x_t, y_t) + \sum_{x_d \in (X_d)} \beta \mathcal{L}_{KL}(f_S, f_T, x_d) \tag{4}$$

where we introduce a new concept: **distillation dataset** $X_d$ is a set of samples on which the knowledge is transferred from a teacher to a student. The first term in the right hand side of (4) is supervised while the second term is unsupervised. It is obvious that the widely used objective (1) is a special case of (4) when $X_d$ is set to $X_t$.

## 4 WHY SMALL STUDENTS UNDERPERFORM LARGE TEACHERS?

In this part, we systematically analyze the reason for the performance gap between students and teachers in KD based model compression. We first introduce several definitions.

**Definition 4.1** *Memorization Error (ME): For a given task with data distribution $P(X, Y)$, ME measures the degree of a student $f_S$ fitting the outputs of a teacher $f_T$ over the data distribution:*

$$E(f_S, f_T, P) = \mathop{\mathbb{E}}_{x \sim P(X)} M(f_T(x), f_S(x)) \tag{5}$$

where $M$ denotes a distance metric such as KL-divergence or mean square error. When ME is 0, it means that the student can completely memorize the outputs of the teacher over the data distribution. In this paper, we take KL-divergence as $M$.

**Definition 4.2** *Capable Students (CSTs) and Incapable Students (ISTs): network $f_S$ with parameters $\Theta_S$ is a CST of teacher $f_T$ if there exists $\Theta_S$ such that $E(f_S, f_T, P)$=0, otherwise, it is an IST.*

Obviously, a CST is able to fully fit the teacher outputs over data distribution $P(X, Y)$. In contrast, an IST does not have the capacity to fit the teacher. For ISTs, the common belief holds that the student-teacher capacity gap causes the performance gap. For example, we cannot expect a two-layer neural network with 1000 parameters to fit the outputs of ResNet-101 with 1.7M parameters on

Table 1: ME of different networks on CIFAR-10, CIFAR-100, and Tiny ImageNet

| Teacher
Student | WRN-40-2
WRN-16-2 | VGG-13
VGG-8 | ResNet32×4
ResNet8×4 | ResNet-110
VGG-8 | ResNet32×4
ShuffleNetV2 | VGG-13
SN2 | VGG-13
SN3 |
|---|---|---|---|---|---|---|---|
| CIFAR-10 | 0.0 | 0.0 | 0.0 | 0.0 | 0.0 | 1.7 | 0.1 |
| CIFAR-100 | 0.0 | 0.0 | 0.0 | 0.0 | 0.0 | 2.4 | 0.3 |
| Tiny ImageNet | 0.0 | 0.0 | 0.0 | 0.0 | 0.0 | 4.2 | 1.9 |

Table 2: Simulation results on CIFAR-100 in terms of test accuracy (%)

| Teacher
Student | ResNet32×4
ResNet8×4 | WRN-40-2
WRN-16-2 | VGG-13
VGG-8 | ResNet32×4
ShuffleNetV2 | VGG-13
SN2 | VGG-13
SN3 |
|---|---|---|---|---|---|---|
| Teacher
Vanilla Student | 79.52
72.50 | 75.81
73.26 | 74.97
70.36 | 79.52
71.82 | 74.97
26.29 | 74.97
55.31 |
| Student Type | CST | CST | CST | CST | IST | IST |
| KD
Simulation KD | 73.33
**79.91** | 74.92
**78.46** | 72.98
**77.99** | 72.14
**81.64** | 26.04
25.58 | 55.32
57.50 |

CIFAR-100. However, in the current SOTA approaches and applications, the commonly used students are modern neural network architectures, such as ResNet-20, ResNet-8×4, VGG-8, and WRN-40-1. We empirically show that these models are CSTs on commonly used benchmark datasets.

To check whether student $f_S$ is a CST of teacher $f_T$ on a task, we minimize ME to check whether $E(f_S, f_T, P)$ can achieve 0. However, in practice, it is impossible to calculate $E(f_S, f_T, P)$ as the data distribution $P$ is typically unknown. Fortunately, we have the access to a set of training data $(X_t, Y_t)$. With the training data, we approximate ME $E(f_S, f_T, P)$ with the empirical error:

$$E_{em}(f_S, f_T, X_t) = \frac{1}{|X_t|} \sum_{x_t \in X_t} M(f_T(x_t), f_S(x_t)) \qquad (6)$$

For comparison, we also evaluate two small neural networks which are expected to be ISTs, i.e., SN-2 and SN-3 with two and three layers, respectively. We report the ME in Table 1 [3], where we adopt the students and the teachers that share the same architectures (e.g., WRN-40-2 and WRN-16-2) or use different architectures (e.g., ResNet-110 and VGG-8). As expected, the widely used students achieve ME 0.0 on these benckmark datasets, i.e., CIFAR-10, CIFAR-100, and Tiny ImageNet while the small networks (i.e., SN2 and SN3) have large ME (e.g., 2.4 and 4.2), which demonstrates that the widely used students are CSTs. However, as observed in the existing literature, these CSTs underperform the teachers by a significant margin on the test data. This suggests that these students have well captured the knowledge on sparse training data points but have not well captured the local shapes of the teachers within the data distribution.

**Corollary 4.1** *In KD, for CSTs, only fitting the outputs of teachers on sparse training data points cannot enable them to well capture the local, in-distribution shapes of the teachers, thus leading to a performance gap. For ISTs, capacity differences cause the performance gap.*

Proof: We empirically show this by comparing the student performances in the following two settings: (a) setting the distillation dataset to training data points; (b) setting the distillation dataset to real data distribution $P(X)$. As $P(X)$ is typically unknown in practice, we conduct a simulation experiment on CIFAR-100. We suppose that the union of the training dataset and the test dataset in CIFAR-100 can accurately represent the real data distribution for this task. Then we randomly draw data samples from the vicinity around the training data and the test data as the distillation dataset, i.e., $X_d$ in (4). Consequently, the distillation dataset can sufficiently represent the real data sample distribution. Note that in the experiments, we never spy the ground truth of the test samples, since the distillation dataset does not use ground truth as shown in (4). This means that the students are trained without any additional supervision compared with the teachers as training datset $(X_t, Y_t)$ in (4) does not change. As CSTs are able to fully memorize the outputs of the teachers, we expect them to achieve the same

---

[3]The ME values in Table 1 are accurate to 1 decimal place.

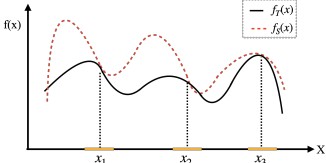 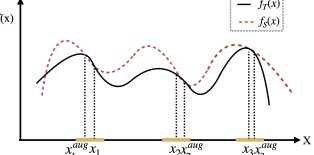 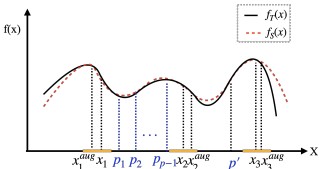

Figure 1: KD with training data    Figure 2: KD with augmentation    Figure 3: KD+

accuracies as or higher accuracies than those of the teachers. In contrast, we expect ISTs to achieve lower accuracies than those of the teachers. Table 2 shows the simulation results. As expected, all the CSTs outperform the teachers in the simulation experiments (i.e., Simulation KD). This is due to the following facts: first, by using the simulated distillation dataset, the distillation objective in (4) makes the CSTs fully capture the knowledge of the teachers within the data distribution; second, the cross-entropy objective in (4) enables the CSTs to learn their own knowledge. Consequently, CSTs contain both the teacher knowledge and the knowledge learned on their own, which results in better performances than those of the teachers. SN2 and SN3 still underperform the teachers in the simulation experiments due to their limited capacities. These results empirically prove Corollary 4.1.

The simulation experiments also suggest a way for CSTs to outperform teachers. That is to sufficiently distill the knowledge in the teachers with a well representative distillation dataset. Unfortunately, it is impossible to have such a distillation datset as the real data sample distribution $P(X)$ is typically unknown in practice. Motivated by this, we propose to go beyond in-distribution distillation.

## 5    GOING BEYOND IN-DISTRIBUTION DISTILLATION

### 5.1    DIFFERENCES BETWEEN SUPERVISED LEARNING AND KNOWLEDGE DISTILLATION

As analyzed above, the reason for the performance gap lies in distillation datasets. Distillation datasets are different from training datsets. As shown in (4), training datsets contain both samples and their ground truth, which are used for *standard supervised learning*. In contrast, distillation datsets only contain samples without ground truth, which are used for *knowledge distillation*. Knowledge distillation and standard supervised learning differ substantially. Standard supervised learning is to learn a function $f$ (e.g., a DNN) for mapping $x$ to $y$ where $(x, y)$ follows real data distribution $P(X, Y)$. The quality of $f$ is constrained by the training data $(X_t, Y_t)$ that we have. In contrast, knowledge distillation is to learn a function (i.e., a student $f_S$) for mapping $x$ to $z$ where $(x, z)$ follows a teacher-defined distribution $Q(X, Z)$ and $Z = \sigma(\frac{f_T(X)}{\tau})$. $Q(X, Z)$ is different from $P(X, Y)$ unless teacher $f_T$ is perfect. The advantage of knowledge distillation is that $Q(X, Z)$ **is more tractable than** $P(X, Y)$, since for given any sample $x$, $f_T$ can always give output $f_T(x)$, and $(x, \sigma(\frac{f_T(x)}{\tau}))$ follows $Q(X, Z)$. Even for out-of-distribution samples, $f_T$ can still output soft labels although these soft labels are semantically meaningless. However, the existing literature ignores this advantage as they only distill the knowledge on sparse training data points.

### 5.2    IMPROVING KD BY FURTHER EXPLORING TEACHER-DEFINED DISTRIBUTION

The simulation experiments demonstrate that only fitting sparse individual data points cannot necessarily enable students to well capture the local shapes of the teacher functions. As shown in Figure 1 where the yellow regions denote real data sample distribution $P(X)$, even if student $f_S$ perfectly fits teacher $f_T$ at each training data point, i.e., $x_1$, $x_2$, and $x_3$, their local shapes near these samples can still be highly different. To mitigate this issue, the typically used strategy is data augmentation Simard et al. (1998) formalized by the *Vicinal Risk Minimization* (VRM) Chapelle et al. (2001) principle. In VRM, human knowledge is necessary to define a vicinity or neighborhood around each training data point. Then, additional new data points can be drawn from the vicinity distribution of the training data. For example, in image classification, it is common to define the vicinity of an image as the set of its random crops after mildly padding and flipping. Nevertheless, data augmentation has its own limitation that a newly generated data point is very close to the original training data point, since they contain almost the identical objective only with different backgrounds caused by padding or cropping. Due to this limitation, as shown in Figure 2, even if the student $f_S$ fits the teacher $f_T$ at all the training data points (i.e., $x_1$, $x_2$, and $x_3$) and the augmented data points (i.e., $x_1^{aug}$, $x_2^{aug}$, and $x_3^{aug}$), their local shapes can still differ substantially.

Table 3: Ablation study on CIFAR-100 in terms of test accuracy (%)

| | Teacher: WRN-40-2, Student: WRN-40-1 | | | | Teacher: ResNet32×4, Student: ShuffleNetV2 | | | |
|---|---|---|---|---|---|---|---|---|
| KD | 73.54±0.20 | | | | 74.45±0.27 | | | |
| | p=2 | p=3 | p=5 | p=10 | p=2 | p=3 | p=5 | p=10 |
| r=2:1 | 75.10±0.13 | 74.79±0.17 | 74.72±0.19 | 74.68±0.24 | 75.72±0.32 | 76.03±0.15 | 75.77±0.12 | 76.03±0.17 |
| r=1:1 | 75.21±0.14 | **75.35±0.16** | 75.13±0.17 | 74.72±0.31 | 76.80±0.08 | **77.22±0.21** | 76.42±0.27 | 76.44±0.29 |
| r=1:2 | 75.00±0.11 | 75.25±0.12 | 74.76±0.32 | 74.76±0.41 | 76.50±0.07 | 77.21±0.14 | 76.57±0.17 | 76.01±0.38 |
| r=1:5 | 74.40±0.18 | 75.15±0.14 | 74.16±0.42 | 74.25±0.34 | 75.35±0.34 | 76.02±0.37 | 75.17±0.35 | 75.50±0.32 |

To address the above issue, we propose KD+ that regularizes KD by enforcing students to mimic the behavior of teachers on the region between two training (or augmented) samples. As shown in Figure 3, KD+ first defines $p-1$ points (i.e., $p_1, p_2, ..., p_{p-1}$) that evenly divide the region between two training samples (i.e., $x_1$ and $x_2$) into $p$ pieces. We denote the set of $p_1, p_2, ...,$ and $p_{p-1}$ by P. P contains in-distribution points and out-of-distribution points. KD+ enforces the students to mimic the behavior of the teachers on P, which serves as a data-driven regularizer. KD+ goes beyond in-distribution distillation as it also uses out-of-distribution points in the regulaizer. As seen from Figure 3, the regularizer can make the student better explore and capture the local shape of the teacher. Consequently, the complete objective of KD+ is written as:

$$\mathcal{L}_{KD+} = \mathcal{L}_{KD} + \lambda \sum_{p_i \in P} \mathcal{L}_{KL}(f_S, f_T, p_i) \tag{7}$$

where $\lambda$ is a balancing weight and simply setting $\lambda$ to 1 works pretty well. KD+ is a very concise approach without requiring complex hyperparameter tuning and can sufficiently explore the knowledge in the teacher by using freely obtained in-distribution and out-of-distribution points as a regularizer.

## 6 EXPERIMENTS FOR EVALUATING KD+

In this section, We first conduct ablation study. Then we show that KD+ is superior to KD by (1) comparing KD+ with KD and other SOTA approaches, (2) showing that KD+ is more compatible with these approaches, (3) showing the superiority of KD+ under few-shot setting.

### 6.1 DATASETS, ARCHITECTURES, COMPETITORS, AND HYPER-PARAMETERS

Experiments are conducted on three benchmark datasets. CIFAR-100 (Krizhevsky & Hinton, 2009) has 100 classes with 50k training images and 10k test images. Tiny ImageNet [4] has 200 classes with 100k training images and 10k test images. ImageNet (Deng et al., 2009) has 1000 classes with 1.28M training images and 50k validation images. We use the standard data augmentation strategy for each dataset. We adopt various modern architectures, i.e., ResNet (He et al., 2016), WRN (Zagoruyko & Komodakis, 2016), VGG (Simonyan & Zisserman, 2015), MobileNet (Sandler et al., 2018), and ShuffleNet (Ma et al., 2018). We compare KD+ with KD and several SOTA methods, i.e., FitNet (Romero et al., 2015), AT (Zagoruyko & Komodakis, 2017), SP (Tung & Mori, 2019), FT (Kim et al., 2018), NST (Huang & Wang, 2017), CC (Peng et al., 2019), FSP (Yim et al., 2017), PKT (Passalis & Tefas, 2018), AB (Heo et al., 2019b), VID (Ahn et al., 2019), RKD (Park et al., 2019), CRD (Tian et al., 2020), and SSKD (Xu et al., 2020). For these SOTA approaches, we report the author-reported results, or use author-provided codes and the optimal hyper-parameters if they are publicly available. Otherwise, we use the implementation of Tian et al. (2020).

We follow KD and set $\alpha$, $\beta$, $\lambda$ and $\tau$ to 0.1, 0.9, 1, and 4, respectively, on all the datasets except on ImageNet where we follow the existing literature to set $\alpha = 1$ and $\tau = 3$. We have trained all the networks for 240, 100, and 120 epochs with SGD with momentum 0.9 on CIFAR-100, Tiny ImageNet, and ImageNet, respectively. We set the initial learning rate to 0.05 for for ResNet, WRN, and VGG, and 0.01 for MobileNet and ShuffleNet. On CIFAR, the learning rate is divided by 10 every 30 epochs after the first 150 epochs. On Tiny ImageNet, the learning rate is divided by 5 every 30 epochs. On ImageNet, the learning rate is initilized to 0.1 and is divided by 10 every 30 epochs. More implementation details are reported in Appendix A.

---

[4]https://tiny-imagenet.herokuapp.com

Table 4: Test accuracy on CIFAR-100. Underline denotes that students match or outperform teachers.

| Teacher
Student | VGG-13
VGG-8 | ResNet32×4
ResNet8×4 | WRN-40-2
WRN-40-1 | ResNet-110
ResNet-32 | WRN-40-2
VGG-8 | ResNet32×4
ShuffleNetV2 |
|---|---|---|---|---|---|---|
| Teacher | 74.64 | 79.42 | 75.61 | 74.31 | 75.61 | 79.42 |
| Vanilla Student | 70.36 | 72.50 | 71.98 | 71.14 | 70.36 | 71.86 |
| KD | 72.98±0.19 | 73.33±0.25 | 73.54±0.20 | 73.08±0.18 | 73.51±0.17 | 74.45±0.27 |
| KD+ | **75.05±0.24** | **76.19±0.19** | **75.35±0.16** | **74.22±0.21** | **75.47±0.13** | **77.22±0.21** |
| FitNet | 71.02±0.31 | 73.50±0.28 | 72.24±0.24 | 71.06±0.13 | 71.14±0.17 | 73.54±0.22 |
| AT | 71.43±0.09 | 73.44±0.19 | 72.77±0.10 | 72.31±0.08 | 70.30±0.21 | 72.73±0.09 |
| SP | 72.68±0.19 | 72.94±0.23 | 72.43±0.27 | 72.69±0.41 | 73.12±0.18 | 74.56±0.22 |
| CC | 70.71±0.24 | 72.97±0.17 | 72.21±0.25 | 71.48±0.21 | 70.64±0.20 | 71.29±0.38 |
| VID | 71.23±0.06 | 73.09±0.21 | 73.30±0.13 | 72.61±0.28 | 71.86±0.23 | 73.40±0.17 |
| RKD | 71.48±0.05 | 71.90±0.11 | 72.22±0.20 | 71.82±0.34 | 71.00±0.19 | 73.21±0.28 |
| PKT | 72.88±0.09 | 73.64±0.18 | 73.45±0.19 | 72.61±0.17 | 72.74±0.42 | 74.69±0.34 |
| AB | 70.94±0.18 | 73.17±0.31 | 72.38±0.31 | 70.98±0.39 | 72.21±0.41 | 74.31±0.11 |
| FT | 70.58±0.08 | 72.86±0.12 | 71.59±0.15 | 72.37±0.31 | 68.33±0.22 | 72.50±0.15 |
| CRD | 73.94±0.22 | 75.51±0.18 | 74.14±0.22 | 73.48±0.13 | 74.08±0.20 | 75.65±0.10 |
| NST | 71.53±0.13 | 73.30±0.25 | 72.24±0.22 | 71.96±0.07 | 69.56±0.24 | 74.68±0.26 |

Table 5: Test accuracies (%) on Tiny ImageNet.

| Teacher | Student | KD | KD+ | FitNet | AT | CC | SP | VID | RKD | CRD | PKT | AB |
|---|---|---|---|---|---|---|---|---|---|---|---|---|
| VGG-13
(61.62) | VGG-8
(55.46) | 60.21
±0.19 | **62.20**
**±0.11** | 55.26
±0.20 | 56.82
±0.46 | 54.14
±0.19 | 56.99
±0.42 | 54.57
±0.26 | 56.60
±0.13 | 59.95
±0.23 | 56.36
±0.17 | 55.41
±0.36 |
| WRN-40-2
(61.84) | WRN-40-1
(55.39) | 56.25
±0.15 | **57.65**
**±0.25** | 55.41
±0.31 | 55.84
±0.41 | 55.10
±0.43 | 54.09
±0.26 | 56.07
±0.23 | 55.37
±0.29 | 56.75
±0.33 | 56.31
±0.22 | 55.76
±0.26 |

Table 6: Comparison results on ImageNet

| | Teacher | Student | KD | KD+ | AT | CRD | SP | CC |
|---|---|---|---|---|---|---|---|---|
| TOP-1 | 73.31 | 69.75 | 70.66 | **71.81** | 70.70 | 71.17 | 70.22 | 69.96 |
| TOP-5 | 91.42 | 89.07 | 89.88 | **90.72** | 90.00 | 90.13 | 89.80 | 89.17 |

## 6.2 ABLATION STUDY AND OUT-OF-DISTRIBUTION DISTILLATION

We investigate how the performance varies with the values of $p$. We also check how the performance varies with the number of points used in the regularizer of KD+ as P contains much more samples than the training dataset. We use $r$ to denote the ratio of the number of training samples to the number of samples used in the regularizer. Table 3 reports the results of KD+ with different $p$ and $r$. The value of $p$ determines what points are included in the regularzier of KD+. When $p=2$, it means that we only use the middle points between two training (or augmented) samples in the regularizer. These middle points have a high probability of being out-of-distribution as they do not belong to any predefined classes. As seen from Table 3, by distilling on these middle points (i.e., $p=2$) as a regularizer, KD+ outperforms KD significantly (e.g., from 73.54 to 75.21 on WRN-40-1), which demonstrates that the out-of-distribution samples can be beneficial to knowledge distillation. Note that not all out-of-distribution samples are useful(e.g., randomly generated samples from normal distribution are harmful as shown in Appendix D). The reason for the usefulness of these middle points may be that these points are not far from the real data distribution as they share some statistics with the training data (e.g., the mean, the variance, and the relation among data dimensions). We also notice that the performance of KD+ is not sensitive to the values of $r$ and $p$. The best performance is achieved when $r=1:1$ and $p=3$. Thus, we simply set $r=1:1$ and $p=3$ in the rest of the experiments.

## 6.3 COMPARISON WITH KD AND SOTA APPROACHES

Table 4 summarizes the comparison results on CIFAR-100. We have the following observations. First, there is an obvious performance gap between the students and the teachers for the existing approaches (e.g., KD, FitNet, and AT). Second, with a simple regularizer, KD+ substantially reduces the performance gap on all the six teacher-student pairs, and even matches or outperforms the teachers on four pairs (that are denoted by underline). On the other two teacher-student pairs, although the students still underperform the teachers, the performance gap is largely reduced by KD+. Note that

Table 7: Compatibility performances on CIFAR-100

| Teacher
Student | ResNet32×4
ResNet8×4 | VGG-13
VGG-8 | ResNet-50
VGG-8 | ResNet32×4
ShuffleNetV2 | ResNet-110
ResNet-20 | WRN-40-2
WRN-16-2 |
|---|---|---|---|---|---|---|
| FitNet+KD | 74.66±0.26 | 73.22±0.21 | 73.24±0.27 | 75.15±0.19 | 70.67±0.21 | 75.12±0.33 |
| FitNet+KD+ | **76.22±0.23** | **74.68±0.16** | **75.48±0.22** | **77.90±0.28** | **71.27±0.26** | **75.89±0.15** |
| AT+KD | 74.53±0.18 | 73.48±0.19 | 74.01±0.25 | 75.39±0.29 | 70.97±0.17 | 75.32±0.15 |
| AT+KD+ | **75.41±0.17** | **74.08±0.25** | **74.93±0.13** | **75.97±0.31** | **71.23±0.23** | **75.87±0.18** |
| SP+KD | 74.02±0.24 | 73.49±0.19 | 73.52±0.25 | 74.88±0.16 | 71.02±0.22 | 74.98±0.28 |
| SP+KD+ | **75.34±0.28** | **74.77±0.16** | **75.22±0.23** | **76.24±0.18** | **71.21±0.24** | **75.23±0.18** |
| CC+KD | 74.21±0.26 | 73.04±0.15 | 73.48±0.29 | 74.71±0.21 | 70.88±0.20 | 75.09±0.23 |
| CC+KD+ | **76.28±0.23** | **74.45±0.22** | **75.33±0.27** | **77.35±0.19** | **71.32±0.27** | **75.88±0.26** |
| VID+KD | 74.56±0.10 | 73.19±0.20 | 73.46±0.25 | 74.85±0.28 | 71.10±0.18 | 75.14±0.15 |
| VID+KD+ | **76.07±0.19** | **75.08±0.24** | **75.63±0.17** | **77.00±0.29** | **71.48±0.16** | **75.70±0.23** |
| RKD+KD | 73.79±0.18 | 72.97±0.08 | 73.51±0.33 | 74.55±0.23 | 70.77±0.16 | 74.89±0.20 |
| RKD+KD+ | **75.81±0.28** | **74.49±0.12** | **75.58±0.31** | **76.47±0.16** | **71.32±0.17** | **75.20±0.22** |
| PKT+KD | 74.23±0.13 | 73.25±0.21 | 73.61±0.28 | 74.66±0.30 | 70.72±0.24 | 75.33±0.18 |
| PKT+KD+ | **76.12±0.22** | **74.80±0.19** | **75.70±0.15** | **76.63±0.12** | **71.36±0.20** | **75.65±0.29** |
| CRD+KD | 75.46±0.25 | 74.29±0.12 | 74.58±0.27 | 76.05±0.09 | 71.56±0.16 | 75.64±0.21 |
| CRD+KD+ | **76.84±0.19** | **74.74±0.21** | **75.82±0.17** | **76.89±0.12** | **72.00±0.23** | **76.08±0.20** |
| AB+KD | 74.40±0.27 | 73.35±0.20 | 73.65±0.41 | 74.99±0.35 | 70.97±0.19 | 70.27±0.17 |
| AB+KD+ | **75.95±0.33** | **74.92±0.21** | **76.11±0.18** | **77.85±0.16** | **71.75±0.27** | **71.35±0.29** |
| NST+KD | 74.28±0.22 | 73.33±0.15 | 71.74±0.29 | 75.24±0.40 | 71.01±0.24 | 74.67±0.26 |
| NST+KD+ | **75.60±0.19** | **74.53±0.23** | **73.85±0.17** | **77.67±0.27** | **71.13±0.29** | **75.68±0.32** |
| SSKD | 76.20±0.36 | 75.33±0.27 | 75.76±0.40 | 78.61±0.33 | 71.38±0.26 | 76.04±0.21 |
| SSKD+ | **76.59±0.11** | **75.60±0.21** | **76.01±0.25** | **78.75±0.18** | **71.54±0.20** | **76.34±0.27** |

there is no guarantee for KD+ to make students match or outperform teachers as the regularizer in KD+ cannot fully compensate for the unknown data sample distribution. Third, KD+ consistently outperforms KD and the other SOTA approaches by a large margin across different architectures, which demonstrates the superiority of KD+. Fourth, on the pair of WRN-40-2 and VGG-8, almost all the representation-based approaches (e.g., FitNet and AT) fail to transfer knowledge from the teacher to the student, even underperform the vanilla student. The reason is that WRN-40-2 and VGG-8 have extremely different architectures. Aligning their feature maps hurts the student performance. In contrast, KD+ shows its robustness and superiority in this case, and even enables student VGG-8 to match the performance of teacher WRN-40-2.

Table 5 reports the comparison results on Tiny ImageNet. KD+ beats KD and the other approaches significantly, and even outperforms teacher VGG-13, which demonstrates the effectiveness of KD+.

We further evaluate KD+ on large scale dataset ImageNet. Limited by computation resources, we only adopt one teacher-student pair on ImageNet. We follow CRD and use ResNet-34 and ResNet-18 as the teacher and the student, respectively. As shown in Table 6, KD+ improves the accuracy over KD and the other approaches significantly, which demonstrates the applicability and usefulness of KD+ on large scale datasets. We also notice that there is still an obvious performance gap between the teacher and the student on ImageNet. The reason can be the model capacity difference as we find that ResNet-18 is an IST of ResNet-34 on the large and complex dataset ImageNet.

## 6.4 COMPATIBILITY WITH SOTA APPROACHES

The existing SOTA approaches can be combined with KD to obtain further performance gain. We show that these approaches combined with KD+ are able to obtain more performance gain. As shown in Table 7, the existing approaches when combined with KD+ consistently achieve much better performances than when combined with KD in all the settings where the teachers and the students use similar or different architectures. This demonstrates that KD+ has a better compatibility than KD and the regularizer of going beyond in-distribution distillation also benefits the existing approaches.

Table 8: Test accuracies on CIFAR-100 under few-shot scenario

|  | 60% Training Data | | 40% Training Data | | 20% Training Data | | 10% Training Data | |
|---|---|---|---|---|---|---|---|---|
| Teacher | ResNet32×4 | VGG-13 | ResNet32×4 | VGG-13 | ResNet32×4 | VGG-13 | ResNet32×4 | VGG-13 |
| Student | ResNet8×4 | VGG-8 | ResNet8×4 | VGG-8 | ResNet8×4 | VGG-8 | ResNet8×4 | VGG-8 |
| Teacher | 79.42 | 74.64 | 79.42 | 74.64 | 79.42 | 74.64 | 79.42 | 74.64 |
| Vanilla Student | 68.54 | 65.57 | 64.35 | 61.45 | 54.70 | 52.50 | 42.76 | 39.30 |
| KD | 69.12 | 69.90 | 66.44 | 66.89 | 58.23 | 59.14 | 47.95 | 49.00 |
| KD+ | **73.65** | **72.10** | **70.75** | **70.53** | **65.68** | **64.94** | **57.00** | **56.50** |
| FitNet | 69.61 | 66.60 | 66.97 | 62.06 | 60.18 | 53.57 | 51.46 | 39.89 |
| AT | 69.35 | 67.36 | 66.19 | 65.12 | 57.72 | 58.16 | 44.70 | 47.16 |
| SP | 69.62 | 69.76 | 65.82 | 66.40 | 59.00 | 58.57 | 45.44 | 40.27 |
| CC | 68.37 | 65.37 | 64.26 | 60.60 | 54.68 | 51.27 | 42.74 | 39.16 |
| VID | 69.12 | 67.29 | 65.87 | 62.58 | 57.31 | 54.86 | 44.41 | 41.07 |
| RKD | 67.71 | 66.18 | 63.51 | 62.32 | 52.29 | 52.13 | 39.19 | 39.70 |
| PKT | 70.48 | 69.21 | 66.41 | 65.97 | 59.06 | 58.08 | 43.50 | 41.15 |
| CRD | 71.29 | 70.46 | 68.15 | 66.27 | 59.38 | 57.57 | 48.23 | 46.33 |
| AB | 69.25 | 65.98 | 65.30 | 63.07 | 58.48 | 56.55 | 48.61 | 48.27 |
| FT | 67.05 | 64.88 | 63.38 | 60.37 | 53.85 | 50.42 | 39.55 | 38.10 |
| NST | 69.87 | 67.12 | 66.24 | 63.56 | 60.27 | 56.63 | 51.91 | 47.44 |

## 6.5 FEW-SHOT SCENARIO

In reality, it can happen that a powerful model is released, but only a few data samples are publicly accessible due to the privacy or confidentiality issues. We evaluate KD+ under few-shot scenario where knowledge is transferred from a powerful teacher to a student with limited data. Table 8 presents the comparison results. We observe that KD+ outperforms KD and the other approaches by a large margin in all the cases with 60%, 40%, 20%, and 10% training data available. The superiority of KD+ becomes more obvious under few-shot scenario, e.g., 9.05% accuracy improvement over KD on ResNet8×4 with 10% training data. The reason is that under few-shot scenario, the training data becomes extremely sparse. Corollary 4.1 holds strongly that only fitting sparse data points cannot enable the students to well capture the local shapes of the teachers. KD+ substantially mitigates this issue by using a regularizer to go beyond the sparse in-distribution distillation.

## 7 CONCLUSION

In this paper, we systematically study why students underperform teachers and how students can outperform teachers under KD based model compression. Through designing exploratory experiments, we find that model capacity differences are not necessarily the root reason and the distillation data matters when the student capacity is greater than a threshold. Inspired by this, we propose KD+ which goes beyond in-distribution distillation. Extensive experiments demonstrate that KD+ is superior to KD as it outperforms KD and the other SOTA approaches substantially, is more compatible with the existing approaches, and shows obvious superiority in few-shot scenario.

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

## A   MORE IMPLEMENTATION DETAILS

The code for this work will be released online. Besides the hyper-parameters reported in the paper, below we report more implementation details.

We adopt the standard preprocessing and data augmentation strategies for each dataset. Each image is preprocessed by subtracting the mean of the whole training set and dividing it by the standard deviation. We use the standard data augmentation strategy, i.e., randomly flipping horizontally, padding 4 pixels for CIFAR (8 pixels for Tiny ImageNet), and then cropping to $32 \times 32$ for CIFAR ($64 \times 64$ for Tiny ImageNet). On ImageNet, we use the widely used scale and aspect ratio augmentation strategy.

On exploratory experiments, the architectures of SN2 and SN3 are Conv(128)-BN-AvgPooling(32)-FC and Conv(128)-BN-ReLU-Covn(256)-BN-ReLU-AvgPooling(16)-FC, respectively.

Following KD, we set $\alpha$, $\beta$, $\lambda$ and $\tau$ to 0.1, 0.9, 1, and 4, respectively. On CIFAR-100, we have trained all the networks for 240 epochs with SGD with momentum 0.9 and batch size 64. On Tiny ImageNet, all the networks are trained with SGD with momentum 0.9 for 100 epochs with batch size 64. On ImageNet, we have the network for 120 epochs with SGD with momentum 0.9 and batch size 256.

For the SOTA approaches, their objective is a combination of the regular cross-entropy loss and a distillation loss:

$$\mathcal{L} = \mathcal{L}_{CE} + c\mathcal{L}_{distill} \tag{8}$$

where $c$ is a weight for balancing the two terms. We report the author-reported results, or use author-provided codes and the optimal hyper-parameters from the original papers if they are publicly available. Otherwise, we use the implementation of Tian et al. (2020). Specifically, the hyper-parameters for each method are: (1) FitNet: $c = 100$; (2) AT: $c = 1000$; (3) SP: $c = 3000$; (4) CC: $c = 0.02$; (5) VID: $c = 1$; (6) RKD: $c_1 = 25$ for the distance metric and $c_2 = 50$ for the angle metric; both terms are combined following the original paper; (7) PKT: $c = 30000$; (8) AB: $c = 0$; distillation happens in the pre-training stage where only distillation objective is used; (9) FT: $c = 500$; (10) CRD: $c = 0.8$; (11) FSP: $c = 0$; distillation happens in the pre-training stage where only distillation objective is used; (12) NST: $c = 50$; (13) the KD objective is (1); $\alpha$, $\beta$ and $\tau$ is set to 0.1, 0.9, and 4, respectively.

For compatibility experiments, KD+ is combined with the existing SOTA approaches. The objective is written as:

$$\mathcal{L}_{CompKD+} = \mathcal{L}_{KD+} + c\mathcal{L}_{distill} \tag{9}$$

The values of $c$ have been reported above.

When the state-of-the-art approaches are combined with KD, the objective is:

$$\mathcal{L}_{CompKD} = \mathcal{L}_{KD} + c\mathcal{L}_{distill} \tag{10}$$

For all the experiments, we report the last epoch test accuracy over 3 runs.

## B  TEACHER-STUDENT SHAPE DIFFERENCES

As stated in Corollary 4.1, only fitting the teacher outputs at sparse data points cannot enable students to well capture the local, in-distribution shapes of teachers. In this part, we show that the students trained with KD+ can better capture the local shapes of the teachers than those trained with KD. The local shape of a function can be represented by a set of pairs $(x, y)$ where $x$ is the input and $y$ is the output of the function. To measure the shape difference, we report the average mean square **student-teacher output logit differences (S-T DIFs)** by using test data as inputs. As

Table 9: S-T DIFs (Shape differences) on CIFAR-100

| Teacher
Student | ResNet32×4
ResNet8×4 | WRN-40-2
WRN-16-2 | WRN-40-2
WRN-40-1 | VGG-13
VGG-8 |
|---|---|---|---|---|
| KD | 2.81 | 2.74 | 2.94 | 1.77 |
| KD+ | **1.45** | **2.02** | **2.10** | **1.11** |

shown in Table 9, S-T DIFs of KD+ are consistently smaller than those of KD, which demonstrates that the student shapes of KD+ are closer to the teacher shapes and indicates that the regularizer benefits the students in capturing the local shpaes of the teachers.

## C  COMPARISON WITH THE REGULARIZER OF INJECTING NOISE TO INPUTS

KD+ goes beyond in-distribution distillation by using a data-driven regularizer. We compare the regularizer in KD+ with the regularizer of injecting small noise to inputs. Intuitively, distilling on noise-injected samples can also explore more knowledge in the teacher. We call this method NoiseKD. We compare KD+ with NoiseKD. We grid search the best hyperparameter for NoiseKD by using different levels of Gaussian noise, i.e., $\mathcal{N}(0, 0.1)$, $\mathcal{N}(0, 0.05)$, $\mathcal{N}(0, 0.01)$, and $\mathcal{N}(0, 0.005)$. Table **??** reports the comparison results. It is observed that when the noise in NoiseKD is large (e.g., $\mathcal{N}(0, 0.1)$ and $\mathcal{N}(0, 0.05)$), NoiseKD even underperforms KD, which indicates that large noise is harmful for knowledge distillation. When noise is relatively small (e.g., $\mathcal{N}(0, 0.01)$), NoiseKD slightly improves the performances over KD, which indicates that small noise is useful for knowledge distillation. We also see that KD+ consistently outperforms NoiseKD with different levels of noise as regularzers, which demonstrates the superiority of the proposed regularizer.

## D  NOT ALL OUT-OF-DISTRIBUTION SAMPLES ARE USEFUL

In KD+, when $p = 2$, the regularizer almost only uses out-of-distribution samples as the middle points of two samples do not belong to any predefined class. The experimental results in Table 3 have shown that by distillation on these out-of-distribution samples as a regularizer, the student performance is improved substantially. Here, we show that not all out-of-distribution samples are useful for knowledge distillation. We randomly draw image-size noise from a normal distribution. And then we distill on these randomly generated noisy samples as a regularizer for KD (we denote this method by NoiseRegKD). The results are reported in Table 10. It is not surprising that the performances drop significantly, e.g., from 72.98 to 6.59 on VGG-8. This indicates that the out-of-distribution samples far from the real data distribution are harmful for knowledge distillation.

## E  LEARNING DATA DISTRIBUTION WITH GENERATIVE MODELS

As stated in Corollary 4.1, only fitting the teacher outputs at sparse data points cannot enable students to well capture the local, in-distribution shapes of teachers. One natural idea is to use generative adversarial networks (GAN) Goodfellow et al. (2014); Arjovsky et al. (2017); Liu et al. (2018) to learn the data distribution and then use the generator to generate fake data for knowledge distillation. However, there are two issues: first, training GAN is computationally expensive especially for large

Table 10: Out-of-distribution distillation as regularizes on CIFAR-100

| Teacher
Student | VGG-13
VGG-8 | WRN-40-2
WRN-16-2 | ResNet32×4
ShuffleNetV2 | ResNet32×4
ResNet8×4 |
|---|---|---|---|---|
| Teacher
Vanilla Student | 74.64
70.36 | 75.61
73.26 | 75.61
71.98 | 79.42
72.50 |
| KD
NoiseRegKD | 72.98
6.59 | 74.92
6.32 | 73.54
16.35 | 73.33
3.26 |

Table 11: KD with GAN as a regularize on CIFAR-10

| Teacher
Student | WRN-40-2
WRN-16-1 | VGG-13
MobileNetV2 |
|---|---|---|
| Teacher
Vanilla Student | 94.80
91.32 | 93.65
89.19 |
| KD
KD-GAN
KD+ | 91.77
92.23
**93.15** | 89.21
89.25
**90.51** |

Table 12: Compatibility performances on CIFAR-100 under few-shot scenario

| | 60% Training Data | | 40% Training Data | | 10% Training Data | |
|---|---|---|---|---|---|---|
| Teacher
Student | ResNet32×4
ResNet8×4 | VGG-13
VGG-8 | ResNet32×4
ResNet8×4 | VGG-13
VGG-8 | ResNet32×4
ResNet8×4 | VGG-13
VGG-8 |
| Teacher
Vanilla Student | 79.42
68.54 | 74.64
65.57 | 79.42
64.35 | 74.64
61.45 | 79.42
42.76 | 74.64
39.30 |
| FitNet+KD
FitNet+KD+ | 72.09
**74.57** | 69.21
**72.23** | 69.64
**72.34** | 65.94
**69.94** | 54.72
**62.43** | 46.30
**56.43** |
| AT+KD
AT+KD+ | 71.54
**73.15** | 70.61
**71.88** | 68.01
**70.47** | 68.22
**69.58** | 50.22
**56.20** | 53.91
**57.08** |
| SP+KD
SP+KD+ | 70.45
**72.42** | 69.70
**72.03** | 67.22
**70.42** | 66.73
**70.28** | 49.94
**55.82** | 45.62
**54.42** |
| CC+KD
CC+KD+ | 70.67
**73.71** | 69.38
**72.70** | 66.81
**70.72** | 66.27
**70.53** | 48.54
**57.00** | 48.37
**57.32** |
| VID+KD
VID+KD+ | 70.00
**73.86** | 69.54
**72.59** | 67.65
**71.25** | 66.42
**70.76** | 47.84
**57.38** | 46.81
**56.98** |
| RKD+KD
RKD+KD+ | 70.33
**72.85** | 69.74
**72.49** | 66.63
**70.47** | 66.15
**69.79** | 46.43
**55.58** | 48.20
**56.12** |
| PKT+KD
PKT+KD+ | 70.98
**73.21** | 70.18
**72.19** | 67.43
**70.83** | 66.13
**70.56** | 48.78
**56.20** | 47.88
**55.63** |
| CRD+KD
CRD+KD+ | 71.97
**73.65** | 70.74
**72.51** | 68.83
**70.29** | 66.84
**69.14** | 48.58
**54.89** | 47.94
**55.84** |
| AB+KD
AB+KD+ | 70.04
**73.76** | 69.95
**73.67** | 67.75
**71.48** | 67.47
**71.77** | 53.76
**60.36** | 60.69
**65.09** |
| NST+KD
NST+KD+ | 71.15
**72.57** | 69.36
**72.20** | 67.92
**71.37** | 66.87
**71.00** | 55.05
**60.28** | 52.49
**59.12** |

datasets (e.g., ImageNet) while KD+ can use freely obtained in-distribution and out-of-distribution points; second, the diversity and quality of the generated fake data from GAN are highly limited by sparse training data, which means that it cannot accurately learn the real data sample distribution,

Table 13: Training time of KD and KD+

| Teacher | VGG-13 | ResNet32×4 | WRN-40-2 |
| Student | VGG-8 | ResNet8×4 | WRN-40-1 |
| --- | --- | --- | --- |
| KD | 7.32s/epoch | 17.21s/epoch | 15.62s/epoch |
| KD+ | 13.32s/epoch | 32.18s/epoch | 28.40s/epoch |

Table 14: Comparison between baselines and baselines+

| Teacher | VGG-13 | WRN-40-2 |
| Student | VGG-8 | WRN-16-2 |
| --- | --- | --- |
| FitNet | 71.02 | 72.24 |
| FitNet+ | **74.15** | **75.09** |
| AT | 71.43 | 72.77 |
| AT+ | **73.55** | **74.81** |
| SP | 72.68 | 72.43 |
| SP+ | **73.81** | **74.59** |
| CC | 70.71 | 72.21 |
| CC+ | **74.21** | **74.99** |
| PKT | 72.88 | 73.45 |
| PKT+ | **74.39** | **75.01** |

just like we cannot obtain 100% test accuracy by training a deep neural network on the training data samples and their ground truth of CIFAR-100.

We conduct an exploratory experiment by using GAN to learn the data sample distribution on CIFAR-10 Krizhevsky & Hinton (2009) as GAN can easily converge on CIFAR-10. And then we distill on the generated fake data as a regularizer for KD. The Results are reported in Table 11. It is obsrved that the GAN regularizer (i.e., KD-GAN) improves the performances over KD, but it underperforms KD+ substantially. This indicates that GAN can generate some useful fake samples for knowledge distillation, but the diversity and usefulness of these samples are highly constrained by the training data. As it is almost impossible to learn the real data sample distribution from sparse training data points, KD+ compensates this by going beyond in-distribution distillation and thus beats KD and the other approaches by a large margin.

## F  COMPATIBILITY WITH SOTA APPROACHES UNDER FEW-SHOT SCENARIO

In this part, we report the compatibility of KD+ with the existing approaches under few-shot scenario as this case can happen in reality where only a few samples are available due to the privacy or confidentiality issues. The comparison results are reported in Table 12. It is observed that the existing approaches when combined with KD+ obtain much better performances than when combined with KD. Moreover, the overall accuracy improvement becomes larger when less training data samples are available. The reason is that when the training data become extremely sparse, Corollary 4.1 holds strongly that only fitting sparse data points cannot enable the students to well capture the local shapes of the teachers. KD+ substantially mitigates this issue by using a regularizer to go beyond the sparse in-distribution distillation.

## G  TRAINING TIME OF KD AND KD+

As KD+ explores more knowledge in the teacher by going beyond in-distribution distillation, it is more computationally expensive than KD. We report the training data on CIFAR-100 with GPU RTX 2080Ti. Both KD and KD+ are trained for 240 epochs. The training time is reported in Table 13.

## H    COMPARISON RESULTS BETWEEN BASELINES AND BASELINES+

To further explore the performances of the proposed approaches on different distillation methods, we compare baselines with baselines+. Baselines+ are obtained by using the P points to assist the baselines (note that the KD+ objective is not included). The comparison results are reported in Table 14. We observe that the beselines+ consistently outperform the baselines by a large margin (e.g., 3.13% accuracy improvement from FitNet to FitNet+), which demonstrates the generalization and effectiveness of the proposed strategy across different distillation approaches.

