# OpenReview forum: "Can Students Outperform Teachers in Knowledge Distillation based Model Compression?"
_ICLR.cc/2021/Conference — Reject_

### Official Review · AnonReviewer3 · 2020-10-21
**Official Blind Review #3**

**Rating:** 6
**Confidence:** 4

**Review:**

The paper studies knowledge distillation. In particular, it tries to disentangle the effect of student model capacity and distillation dataset on the performance of the student. The paper goes on to present KD+, a knowledge distillation approach that goes beyond in-distribution data. Experiments on multiple image recognition models and datasets show that KD+ outperforms KD consistently.

The study is very interesting in how it tries to characterize the impact of  student capacity and distillation data on the performance of  the distilled student. The ablations studies are interesting and shed light on several angels including few-shot learning scenarios, different algorithms for distillation, etc.

There are some important questions though that arise:

- Is the effect of improved student performance coming from having out-of-distribution data or simply more data for distillation? This also relates to the bigger gains with few-shot learning settings.

- When using p=2 for data augmentation (adding a middle point between two samples), are classes considered in any way? In other words, could it be that the model is benefiting from adding harder examples to the distillation set? Similar observations were made in self-training where there has been a lot of work on leveraging how hard an example is, or how  confident a model is to select samples for self-training?

- The gains seem to vary given the distillation algorithm, with bigger gain when using basic KD. Could other distillation algorithms that try to align the representations of the student and teacher by using different criteria be accomplishing similar generalization effect to the proposed approach

---

> ### Author Response · Authors · 2020-11-12
> **Response to Reviewer #3**
>
>
> R1: Is the effect of improved student performance coming from having out-of-distribution data or simply more data for distillation ...\
> A1: KD+ benefits from both. About out-of-distribution data: the ablation study in Table 3 demonstrates that out-of-distribution data are beneficial to KD. However, not all out-of-distribution data are beneficial as shown in Appendix D. About more data: as KD+ does both in-distribution and out-of-distribution distillation, it uses more samples. As shown in Table 3, the performance first increases and then drops a little as the KD+ uses more and more samples in the P set. This means that KD+ benefits from more samples from P when the number is less a threshold.
>
> R2: When using p=2 for data augmentation … are classes considered in any way? In other words, could it be that the model is benefiting from adding harder examples to the distillation set …\
> A2: We did not consider the classes as KD+ aims to enforce the student to fit the local shape of the teacher. We conduct experiments to explore the effect of harder examples. We denote the strategy by Hard_KD+ that only uses harder samples in the P set. The harder samples are selected by the following strategy: (1) randomly sample a batch of data points from the P set. (2) rank these samples based on the entropy of the output probability of the students on these data points. (3) select the top 50% samples as harder samples, since their entropy is larger than those of the others. We compare KD+ with Hard_KD+.\
> ——————————————————————\
> Teacher&nbsp;&nbsp;&nbsp;&nbsp;VGG-13&nbsp;&nbsp;&nbsp;&nbsp;ResNet32$\times$4&nbsp;&nbsp;&nbsp;&nbsp;WRN-40-2\
> Student&nbsp;&nbsp;&nbsp;&nbsp;VGG-18&nbsp;&nbsp;&nbsp;&nbsp;ResNet8$\times$4&nbsp;&nbsp;&nbsp;&nbsp;&nbsp;&nbsp;WRN-40-1\
> ——————————————————————\
> Hard_KD&nbsp;&nbsp;&nbsp;&nbsp;74.98&nbsp;&nbsp;&nbsp;&nbsp;&nbsp;&nbsp;&nbsp;&nbsp;&nbsp;76.21&nbsp;&nbsp;&nbsp;&nbsp;&nbsp;&nbsp;&nbsp;&nbsp;&nbsp;&nbsp;&nbsp;&nbsp;&nbsp;&nbsp;&nbsp;&nbsp;&nbsp;&nbsp;&nbsp;75.15\
> KD+&nbsp;&nbsp;&nbsp;&nbsp;&nbsp;&nbsp;&nbsp;&nbsp;&nbsp;&nbsp;&nbsp;&nbsp;75.05 &nbsp;&nbsp;&nbsp;&nbsp;&nbsp;&nbsp;&nbsp;&nbsp;76.19&nbsp;&nbsp;&nbsp;&nbsp;&nbsp;&nbsp;&nbsp;&nbsp;&nbsp;&nbsp;&nbsp;&nbsp;&nbsp;&nbsp;&nbsp;&nbsp;&nbsp;&nbsp;&nbsp;75.35\
> ——————————————————————\
> It is observed that there is much performance difference between them.
>
> R3: The gains seems … align the representations of the student and teacher by using different criteria be accomplishing similar generalization effect to the proposed approach\
> A3: The other distillation approaches can benefit from the proposed strategy substantially. We report the comparison results on CIFAR-100  as follows, where a baseline+ is obtained by using the P points to assist the baseline (note the KD+ objective is not included). We observe that the beselines+ consistently outperform the baselines by a large margin (e.g., 3.13% accuracy improvement from FitNet to FitNet+), which demonstrates the generalization and effectiveness of the proposed strategy across different distillation approaches. We have added this interesting experiment to Appendix H in the revised paper.\
> ———————————————\
> Teacher&nbsp;&nbsp;&nbsp;&nbsp;VGG-13&nbsp;&nbsp;&nbsp;&nbsp;WRN-40-2\
> Student&nbsp;&nbsp;&nbsp;&nbsp;VGG-8&nbsp;&nbsp;&nbsp;&nbsp;&nbsp;&nbsp;WRN-40-1\
> ———————————————\
> FitNet&nbsp;&nbsp;&nbsp;&nbsp;&nbsp;&nbsp;&nbsp;&nbsp;71.02&nbsp;&nbsp;&nbsp;&nbsp;&nbsp;&nbsp;&nbsp;&nbsp;72.24\
> FitNet+ &nbsp;&nbsp;&nbsp;&nbsp;&nbsp;74.15&nbsp;&nbsp;&nbsp;&nbsp;&nbsp;&nbsp;&nbsp;&nbsp;75.09\
> ———————————————\
> AT&nbsp;&nbsp;&nbsp;&nbsp;&nbsp;&nbsp;&nbsp;&nbsp;&nbsp;&nbsp;&nbsp;&nbsp;&nbsp;&nbsp;71.43&nbsp;&nbsp;&nbsp;&nbsp;&nbsp;&nbsp;&nbsp;&nbsp;72.77\
> AT+&nbsp;&nbsp;&nbsp;&nbsp;&nbsp;&nbsp;&nbsp;&nbsp;&nbsp;&nbsp;&nbsp;&nbsp;73.55&nbsp;&nbsp;&nbsp;&nbsp;&nbsp;&nbsp;&nbsp;&nbsp;74.81\
> ———————————————\
> SP&nbsp;&nbsp;&nbsp;&nbsp;&nbsp;&nbsp;&nbsp;&nbsp;&nbsp;&nbsp;&nbsp;&nbsp;&nbsp;&nbsp;72.68&nbsp;&nbsp;&nbsp;&nbsp;&nbsp;&nbsp;&nbsp;&nbsp;72.43\
> SP+&nbsp;&nbsp;&nbsp;&nbsp;&nbsp;&nbsp;&nbsp;&nbsp;&nbsp;&nbsp;&nbsp;&nbsp;73.81&nbsp;&nbsp;&nbsp;&nbsp;&nbsp;&nbsp;&nbsp;&nbsp;74.59\
> ———————————————\
> CC&nbsp;&nbsp;&nbsp;&nbsp;&nbsp;&nbsp;&nbsp;&nbsp;&nbsp;&nbsp;&nbsp;&nbsp;&nbsp;&nbsp;70.71&nbsp;&nbsp;&nbsp;&nbsp;&nbsp;&nbsp;&nbsp;&nbsp;72.21\
> CC+&nbsp;&nbsp;&nbsp;&nbsp;&nbsp;&nbsp;&nbsp;&nbsp;&nbsp;&nbsp;&nbsp;&nbsp;74.21&nbsp;&nbsp;&nbsp;&nbsp;&nbsp;&nbsp;&nbsp;&nbsp;74.99\
> ———————————————\
> PKT&nbsp;&nbsp;&nbsp;&nbsp;&nbsp;&nbsp;&nbsp;&nbsp;&nbsp;&nbsp;&nbsp; 72.88&nbsp;&nbsp;&nbsp;&nbsp;&nbsp;&nbsp;&nbsp;&nbsp;73.45\
> PKT+&nbsp;&nbsp;&nbsp;&nbsp;&nbsp;&nbsp;&nbsp;&nbsp;&nbsp;&nbsp;74.39&nbsp;&nbsp;&nbsp;&nbsp;&nbsp;&nbsp;&nbsp;&nbsp;75.01\
> ———————————————

---

### Official Review · AnonReviewer4 · 2020-10-27
**Review from AnonReviewer4**

**Rating:** 6
**Confidence:** 5

**Review:**

This paper explores why the large gap existing from the viewpoint of data distribution rather than student model capacity. Base on the exploratory experiments and some analyses, the paper proposes KD+ by sampling out-of-distribution data points and achieve better results than current KD methods on the experiment.

Strengths:
1. The exploratory experiments on out-of-distribution data is interesting, which is meaningful to discuss why the gap between the student and teacher model exists from a new viewpoint of data distribution rather than the student capacity.
2. KD+ achieved promising results.


Some concerns:
1. A recent KD work [3] also achieved promising results on ImageNet by adopting a discriminator to judge the output distribution is from the teacher model or student model, I think it shares a similar idea with this work, but it uses GAN-style learning to make student's output be more like teachers. This paper fails to compare their results with [3] or discuss the difference by using a GAN style to learn the output distribution of the teacher.
2.  KD+ will sample data on the original training data, so it would result in more training data than KD or other methods. How does the training time change? The paper fails to discuss training time changes on more training data.
3. The key point of KD+ is to sample from training data $x$, while it only gives an illustration figure (Figure3) on one dimension data $x$ rather than on real images. Namely, the paper doesn't discuss how to sample $p$ images from two images $x_1$ and $x_2$, by interpolating from $x_1$ and $x_2$ or some other methods?
4. In Table 2, the simulation results of students (CSTs) are better than the teacher models. I want to know what if use the teacher model to teach itself on the same simulation experiments, as the work in Born Again Network [1] or Tf_self [2].  The author would like to consider this improvement as regularisation [3] or some dark knowledge from out-of-distribution?


[1] "Meal v2: Boosting vanilla resnet-50 to 80%+ top-1 accuracy on imagenet without tricks." Shen, et al. arXiv:2009.08453 (2020).

[2] "Born again neural networks." Furlanello et al, ICML 2018.

[3] "Revisiting knowledge distillation via label smoothing regularization.", Yuan et al, CVPR 2020.

---

> ### Author Response · Authors · 2020-11-12
> **Response to Reviewer #4**
>
> R1: A recent KD … This paper fails to compare their results with [3] or discuss the difference by using a GAN style to learn the output distribution of the teacher.\
> A1: This GAN-Style-KD deals with high-level network outputs by using GAN-Style training to align the high-level output distributions of the teacher and the student. KD+ addresses knowledge distillation from data perspective and aims to directly align the local shapes of the teacher and the student by exploring more knowledge between two training samples. We compare KD+ with GNA-Style-KD. The results on CIFAR-100 are reported as follows:\
> ——————————————————————————\
> Teacher&nbsp;&nbsp;&nbsp;&nbsp;&nbsp;&nbsp;&nbsp;&nbsp;&nbsp;&nbsp;&nbsp;&nbsp;VGG-13&nbsp;&nbsp;&nbsp;&nbsp;&nbsp;&nbsp;ResNet32$\times$4&nbsp;&nbsp;&nbsp;&nbsp;&nbsp;WRN-40-2\
> Student&nbsp;&nbsp;&nbsp;&nbsp;&nbsp;&nbsp;&nbsp;&nbsp;&nbsp;&nbsp;&nbsp;&nbsp;&nbsp;VGG-8&nbsp;&nbsp;&nbsp;&nbsp;&nbsp;&nbsp;&nbsp;&nbsp;&nbsp;ResNet8$\times$4&nbsp;&nbsp;&nbsp;&nbsp;&nbsp;&nbsp;WRN-40-1\
> ——————————————————————————\
> KD&nbsp;&nbsp;&nbsp;&nbsp;&nbsp;&nbsp;&nbsp;&nbsp;&nbsp;&nbsp;&nbsp;&nbsp;&nbsp;&nbsp;&nbsp;&nbsp;&nbsp;&nbsp;&nbsp;&nbsp;&nbsp;&nbsp;&nbsp;72.98&nbsp;&nbsp;&nbsp;&nbsp;&nbsp;&nbsp;&nbsp;&nbsp;&nbsp;&nbsp;&nbsp;&nbsp;&nbsp;73.33&nbsp;&nbsp;&nbsp;&nbsp;&nbsp;&nbsp;&nbsp;&nbsp;&nbsp;&nbsp;&nbsp;&nbsp;&nbsp;&nbsp;&nbsp;&nbsp;73.54\
> GAN-Style-KD&nbsp;&nbsp;&nbsp;&nbsp;73.37&nbsp;&nbsp;&nbsp;&nbsp;&nbsp;&nbsp;&nbsp;&nbsp;&nbsp;&nbsp;&nbsp;&nbsp;&nbsp;73.98&nbsp;&nbsp;&nbsp;&nbsp;&nbsp;&nbsp;&nbsp;&nbsp;&nbsp;&nbsp;&nbsp;&nbsp;&nbsp;&nbsp;&nbsp;&nbsp;74.00\
> KD+&nbsp;&nbsp;&nbsp;&nbsp;&nbsp;&nbsp;&nbsp;&nbsp;&nbsp;&nbsp;&nbsp;&nbsp;&nbsp;&nbsp;&nbsp;&nbsp;&nbsp;&nbsp;&nbsp;&nbsp;&nbsp;75.05&nbsp;&nbsp;&nbsp;&nbsp;&nbsp;&nbsp;&nbsp;&nbsp;&nbsp;&nbsp;&nbsp;&nbsp;&nbsp;76.19&nbsp;&nbsp;&nbsp;&nbsp;&nbsp;&nbsp;&nbsp;&nbsp;&nbsp;&nbsp;&nbsp;&nbsp;&nbsp;&nbsp;&nbsp;&nbsp;75.35\
> —————————————————————————\
> It is observed that although GAN-Style training improves the performances over KD, it underperforms KD+ by a large margin, which demonstrates the effectiveness of KD+. On the other hand, GAN can also be used to learn the real data distribution and then we can sample fake samples from the generator in GAN. These faker samples are then added to the distillation dataset. Please refer to Appendix E for the comparison results between this strategy and KD+ and the limitations of this strategy.
>
>
> R2: KD+ will sample data on the original training … The paper fails to discuss training time changes on more training data.\
> A2: Yes, as KD+ explores more knowledge in the teacher by going beyond in-distribution distillation, it is more computationally expensive than KD. We report the training data on CIFAR-100 with GPU RTX 2080Ti. Both KD and KD+ are trained for 240 epochs.  The training time is reported as follows (and we add the training time to Appendix G in the revised paper.): \
> ——————————————————————————\
> Teacher&nbsp;&nbsp;&nbsp;&nbsp;VGG-13&nbsp;&nbsp;&nbsp;&nbsp;&nbsp;&nbsp;&nbsp;&nbsp;ResNet32$\times$4&nbsp;&nbsp;&nbsp;&nbsp;&nbsp;WRN-40-2\
> Student&nbsp;&nbsp;&nbsp;&nbsp;VGG-8&nbsp;&nbsp;&nbsp;&nbsp;&nbsp;&nbsp;&nbsp;&nbsp;&nbsp;&nbsp;&nbsp;ResNet8$\times$4&nbsp;&nbsp;&nbsp;&nbsp;&nbsp;&nbsp;&nbsp;WRN-40-1\
> ——————————————————————————\
> KD&nbsp;&nbsp;&nbsp;&nbsp;&nbsp;&nbsp;&nbsp;&nbsp;7.32s/epoch&nbsp;&nbsp;&nbsp;&nbsp;17.21s/epoch&nbsp;&nbsp;&nbsp;15.62s/epoch\
> KD+&nbsp;&nbsp;&nbsp;&nbsp;&nbsp;13.32s/epoch&nbsp;&nbsp;&nbsp;&nbsp;32.18s/epoch&nbsp;&nbsp;&nbsp;28.40s/epoch\
> ——————————————————————————
>
> R3: The key point of KD+ is … the paper doesn't discuss how to sample images from two images … \
> A3:  As the P points evenly divide the region between two training samples (i.e., x1 and x2) into p pieces, $p_1$ = x1 + 1/p*(x2-x1); $p_2$ = x1 + 2/p*(x2-x1); … $p_i$ = x1 + i/p*(x2-x1).
>
> R4: In Table 2, … what if use the teacher model to teach itself on the same simulation experiments … \
> A4: We conduct this interesting experiment by using two networks on CIFAR-100 and report the results as follows:\
> ———————————————\
> Teacher&nbsp;&nbsp;&nbsp;&nbsp;VGG-13&nbsp;&nbsp;&nbsp;&nbsp;WRN-40-2\
> Student &nbsp;&nbsp;&nbsp;&nbsp;VGG-13&nbsp;&nbsp;&nbsp;&nbsp;WRN-40-2 \
> ——————————————\
> Teacher&nbsp;&nbsp;&nbsp;&nbsp;&nbsp;&nbsp;74.97&nbsp;&nbsp;&nbsp;&nbsp;&nbsp;&nbsp;&nbsp;&nbsp;75.81\
> ——————————————\
> KD&nbsp;&nbsp;&nbsp;&nbsp;&nbsp;&nbsp;&nbsp;&nbsp;&nbsp;&nbsp;&nbsp;&nbsp;&nbsp;&nbsp;76.15&nbsp;&nbsp;&nbsp;&nbsp;&nbsp;&nbsp;&nbsp;&nbsp;77.52\
> Simula-KD&nbsp;80.83&nbsp;&nbsp;&nbsp;&nbsp;&nbsp;&nbsp;&nbsp;80.88\
> ——————————————\
> It is observed that the performances of Simulation KD are much higher than those of the teachers. Note that in this case, there is no model capacity gap, thus KD can also outperform teachers. We believe that the improvement is from label smoothing regularization and instance-to-class similarity information.

---

### Official Review · AnonReviewer1 · 2020-10-29
**The authors insist that this is the first work to explore the root reason for the performance gap in knowledge distillation and the distillation data matters instead of the model capacity. But this paper presents some unconvincing arguments that the designed experiments cannot support firmly. This paper lacks the most important references and the proposed KD+ is not superior to the existing approach.**

**Rating:** 3
**Confidence:** 5

**Review:**

This paper presents an argument that model capacity differences are not necessarily the root reason for the performance gap between the student and the teacher, and the distillation data matters when the student capacity is greater than a threshold. Based on this, the authors develop KD+ to reduce the performance gap between them and enable students to match or outperform their teachers. In addition, this paper designs experiments to confirm the proposed arguments.
However, this paper should be rejected because:
(1)the results of Table 1 cannot confirm the authors’ argument that the widely used students are CSTs. It also depends on the model capacity differences whether a CST is able to fully fit the teacher outputs or not.
(2) some arguments in this paper are unclear and lack verification. For example, the paper states“This suggests that these students have well captured the knowledge on sparse training data points but have not well captured the local shapes of the teachers within the data distribution.” in Definition 4.2. How did you come to this conclusion?
(3) the most important or relevant references are not cited, for example [1] Xu G, Liu Z, Li X, et al. Knowledge Distillation Meets Self-Supervision, 2020. And the proposed KD+ is not superior to the approach of [1], for example, some results in Table 4 and Table 8.
(4) This paper lacks experimental settings and details.
Please refer to the above comments and answer these questions.

---

> ### Author Response · Authors · 2020-11-12
> **Response to the concerns of Reviewer#1**
>
> R1: (1)The results of Table 1 cannot confirm the authors’ argument that the widely used students are CSTs. It also depends on the model capacity differences …\
> A1: As stated in Definition 4.2, if a student can fully memorize the outputs of a teacher on the target task, we call this student as a CST of the teacher. This definition has already considered model capacity differences. Table 1 shows that the widely used students achieve 0.0 memorization error with respect to these teachers on these datasets, which demonstrates that they fully memorize the the outputs of the teachers on these datasets and thus are CSTs of their teachers on these datasets.
>
> R2: Some arguments in this paper are unclear ... states“This suggests that these students have well captured the knowledge on sparse training data points but have not well captured the local shapes of the teachers within the data distribution.” …\
> A2: As real data distribution P(x, y) is typically unknown, we approximate memorization error (ME) by using the training dataset with the empirical error (6). CSTs achieve 0.0 ME on training data as shown in Table 1, which demonstrates that these students have well captured the knowledge on training data points. On the other hand, as observed in the existing literature or Table 4, these CSTs underperform the teachers by a significant margin on the test data, which means that these CTSs have not well captured the local shapes of the teachers on test data. Combining these two points, we stated that ‘’This suggests that these students have well captured the knowledge on sparse training data points but have not well captured the local shapes of the teachers within the data distribution.
>
> R3: the most important or relevant references are not cited, for example [1] Xu G, Liu Z, Li X, et al. Knowledge Distillation Meets Self-Supervision, 2020. And the proposed KD+ is not superior to the approach of [1] …\
> A3: Self-Supervision knowledge distillation (SSKD) addresses KD from a perspective that is different from ours . They use extra self-supervision tasks to assist KD while KD+ addresses KD from data perspective by going beyond in-distribution distillation. SSKD is not a competitor to KD+ as extra tasks and extra (freely obtained) samples are compatible and both beneficial. The real competitor of KD+ is KD as KD+ can be combined with the existing approaches. As stated in the paper, we aims to show KD+ is superior to KD and thus is ready to replace KD by demonstrating that KD+ outperforms KD and is more compatible with the existing approaches (Table 7). Note that the original KD objective is included in SSKD. We use the strategy in KD+ to assist SSKD. The comparison results are reported as follows. We add also the comparison results to Table 7 in the revised paper. It is observed that SSKD+ outperforms SSKD consistently, which demonstrates that extra tasks and extra data are compatible and both helpful. We also notice that the improvement is not large. The reason is that SSKD has already used strong data augmentation (i.e., color dropping, rotation, cropping). The success of SSKD also verifies our argument in Corollary 4.1 in the paper that distillation data matters for CSTs when the model capacity differences are unavoidable.\
> ———————————————————————————————————————\
> Teacher&nbsp;&nbsp;&nbsp;ResNet32$\times$4&nbsp;&nbsp;&nbsp;VGG-13&nbsp;&nbsp;ResNet-50&nbsp;&nbsp;ResNet32$\times$4 &nbsp;&nbsp;&nbsp;ResNet-110&nbsp;&nbsp;&nbsp;WRN-40-2\
> Student&nbsp;&nbsp;&nbsp;ResNet8$\times$4&nbsp;&nbsp;&nbsp;&nbsp;&nbsp;VGG-8&nbsp;&nbsp;&nbsp;&nbsp;&nbsp;&nbsp;VGG-8&nbsp;&nbsp;&nbsp;&nbsp;&nbsp;&nbsp;&nbsp;ShuffleNetV2&nbsp;&nbsp;&nbsp;&nbsp;ResNet-20&nbsp;&nbsp;&nbsp;&nbsp;&nbsp;WRN-16-2\
> ———————————————————————————————————————\
> SSKD&nbsp;&nbsp;&nbsp;&nbsp;&nbsp;&nbsp;&nbsp;&nbsp;&nbsp;&nbsp;76.20&nbsp;&nbsp;&nbsp;&nbsp;&nbsp;&nbsp;&nbsp;&nbsp;&nbsp;&nbsp;&nbsp;&nbsp;&nbsp;&nbsp;75.33&nbsp;&nbsp;&nbsp;&nbsp;&nbsp;&nbsp;&nbsp;75.76&nbsp;&nbsp;&nbsp;&nbsp;&nbsp;&nbsp;&nbsp;&nbsp;&nbsp;&nbsp;&nbsp;&nbsp;78.61&nbsp;&nbsp;&nbsp;&nbsp;&nbsp;&nbsp;&nbsp;&nbsp;&nbsp;&nbsp;&nbsp;&nbsp;&nbsp;&nbsp;&nbsp;&nbsp;&nbsp;&nbsp;71.38&nbsp;&nbsp;&nbsp;&nbsp;&nbsp;&nbsp;&nbsp;&nbsp;&nbsp;&nbsp;&nbsp;&nbsp;&nbsp;76.04\
> SSKD+&nbsp;&nbsp;&nbsp;&nbsp;&nbsp;&nbsp;&nbsp;&nbsp;76.59&nbsp;&nbsp;&nbsp;&nbsp;&nbsp;&nbsp;&nbsp;&nbsp;&nbsp;&nbsp;&nbsp;&nbsp;&nbsp;&nbsp;75.60&nbsp;&nbsp;&nbsp;&nbsp;&nbsp;&nbsp;&nbsp;76.01&nbsp;&nbsp;&nbsp;&nbsp;&nbsp;&nbsp;&nbsp;&nbsp;&nbsp;&nbsp;&nbsp;&nbsp;78.75&nbsp;&nbsp;&nbsp;&nbsp;&nbsp;&nbsp;&nbsp;&nbsp;&nbsp;&nbsp;&nbsp;&nbsp;&nbsp;&nbsp;&nbsp;&nbsp;&nbsp;&nbsp;71.54&nbsp;&nbsp;&nbsp;&nbsp;&nbsp;&nbsp;&nbsp;&nbsp;&nbsp;&nbsp;&nbsp;&nbsp;&nbsp;76.34\
> —————————————————————— —————————————————
>
> R4: This paper lacks experimental settings and details. \
> A4: The hyper-parameters of different approaches and more details are given in Appendix due to the space limitation. We add more details to the revised version of the paper.

---

### Official Review · AnonReviewer2 · 2020-10-30
**Official Blind Review #2**

**Rating:** 5
**Confidence:** 4

**Review:**

This paper investigates why conventional student network is not outperformed to teacher network. Based on several experiments design, this paper argues this is due to the student only fit the local, in-distribution shapes of the network rather than model compacity.  Besides, this paper design a go beyond in-distribution distillation approach to overcome this issue. Overall, I think this paper has some merits, but also some concern need to be addressed.

Pros:
1）This paper is well-written. The motivation and the organization of the paper are easy to follow.
2)   The experiments design to prove the assumption part is good to read and make sense, e.g. Tabel1, Tabel2 CST/ IST

Cons:
1）The first concern is the result of Table 1, I was very supervised the loss of teacher and student is equal to 0. For my experience, the distillation loss in very small, but it still has the same value, which means the student output can not total the same as teacher output.

2）The second concern is about the novelty of KD+, the selection of Out of distribution sample is similar to the technique applied in the current state of art semi-supervised learning methods: 1) mixmatch[1] 2) remixmatch[2]. I suggest the author add these reference and discuss. They also use KD and use middle points.

3) I also concern about the fairness of Table 4, since using the middle points is a regularization technique, like, mixup[3]. Does it fair to compare with KD without mixup.

4) The last one is the student can outperform to teacher has been studied some literature? I remember some of the paper point out this.

Reference:
[1]mixmatch: a holistic approach to semi-supervised learning
[2]remixmatch: semi-supervised learning with distribution alignment and augmentation anchoring
[3]mixup: Beyond Empirical Risk Minimization

---

> ### Author Response · Authors · 2020-11-12
> **Response to the concerns of Reviewer #2**
>
> R1: The first concern is the result of Table 1, I was very supervised the loss of teacher and student is equal to 0. \
> A1:  The ME values reported in Table 1 (e.g., 0.0 and 1.7) are accurate to 1 decimal place so that they look like 0.  The more accurate values on CIFAR-100 are: \
> ————————————————————————————————— \
> Teacher&nbsp;&nbsp;WRN-40-2&nbsp;&nbsp;ResNet324&nbsp;&nbsp;&nbsp;ResNet32$ \times$4&nbsp;&nbsp;VGG-13&nbsp;&nbsp; &nbsp;&nbsp;VGG-13 \
> Student&nbsp;&nbsp;WRN-16-2&nbsp;&nbsp;ResNet8$ \times$4&nbsp;&nbsp;ShuffleNetV2 &nbsp;&nbsp;&nbsp;SN2 &nbsp;&nbsp;&nbsp;&nbsp;&nbsp;&nbsp;&nbsp;&nbsp;SN3 \
> ——————————————————————————————————\
> ME&nbsp;&nbsp;&nbsp;&nbsp;&nbsp;&nbsp;&nbsp;&nbsp;&nbsp;  0.003090&nbsp;&nbsp;&nbsp;&nbsp;&nbsp;0.003961 &nbsp;&nbsp;&nbsp;&nbsp;0.005549 &nbsp;&nbsp;&nbsp;&nbsp; 2.365744&nbsp;&nbsp; 0.262467 \
> ————————————————————————————————— \
> When these values are accurate to 1 decimal place, they will be exactly the values reported in Table 1. We make this point clear in the revised version of this paper.
>
> R2: The second concern is about the novelty of KD+, the selection …\
> A2: We add the discussion about these methods to the related work in the revised paper. As the reviewer pointed out, similar data selection strategies have been used in some other applications, e.g., network calibration, semi-supervised learning, uncertainty estimation, and learning with noise. However, how these freely obtained data and out-of-distribution samples affect the performances of students including CSTs and ISTs in KD based model compression has never been explored.  This paper systematically studies this problem and summarizes and shows the relation between the performances of CSTs, ISTs, teachers and the distillation data. The existing literature usually simply attributes the teacher-student performance gap to the model capacity difference. While the model capacity gap is unavoidable in KD based model compression, we find that distillation data can compensate for this gap for CSTs. The key idea and the motivation between them are arguably different.
>
> R3: I also concern about the fairness of Table 4 …\
> A3: This paper aims to show that distillation data matters when the student capacity is greater than a threshold. Thus, the only difference between KD and KD+ is the distillation dataset. We empirically show that KD+ is superior to KD and even can match the teacher performance, which verifies our argument. Moreover, Mixup aims to enforce local linearity of deep neural networks by linearly interpolating both samples and targets while KD+ aims to make the students better mimic the local shape of the teacher function by exploring the knowledge between two samples.
>
> R4: The last one is the student can outperform to teacher has been studied some literature? I remember some of the paper point out this.\
> A4: The existing literature has observed that a student can outperform a teacher when the student capacity is equal to or greater than the teacher capacity, e.g., Born-Again-Network. However, as stated in the title and the paper, the question studied in the paper is whether students can outperform teachers in KD based model compress, i.e., the student capacity is less than the teacher’s. The work is first to systematically study this question and provide the relation between CSTs, ISTs, teachers, and distillation data.

---

### Author Response · Authors · 2020-11-12
**We appreciate the valuable comments and/or the efforts of the reviewers and the AC. We summarize the changes in the revised paper according to the comments.**

We thank all the four reviewers and the AC for the valuable comments and/or the efforts. We address these concerns in the following and accordingly revise the paper. The revised paper has been uploaded. We summarize the changes in the revised paper as follows:

1. We add all the work mentioned in the reviews and the discussion to the related work.
2. We add a footnote to describe the precision of the results in Table 1.
3. We add the comparison results between SSKD (i.e., Knowledge Distillation meets Self-Supervision)  and SSKD+ to Table 7.
4. We add more experimental setting to Section 6.1 and update the results on ImageNet by following the existing literature to set \alpha to 1 instead of 0.1 in Table 6.
5. We add the training time of KD and KD+ in Appendix G.
6. We add the comparison between baselines and baselines+ to Appendix H where baselines+ means that we use the P set to assist the baselines.

---

### Decision · Program_Chairs · 2021-01-07
**Final Decision**

**Decision:**

Reject

**Comment:**

The paper studies Knowledge Distillation (KD) to better understand the reasons behind the performance gap between student and teacher models. The analysis is done by conducting exploratory experiments. The paper establishes that the distillation data used for training a student can play a critical role in the performance gap apart from the model capacity. Building on this idea, the authors propose a new approach to distillation, KD+, utilizing out-of-distribution data when training a student. Extensive experiments are performed to demonstrate the efficiency of KD+. Overall, the paper studies an interesting problem. The results provide a more in-depth explanation of how the distillation data and model capacity play a role in the performance gap between student and teacher models in KD.

I want to thank the authors for providing the rebuttal and sharing their concerns about the quality of one of the reviews.  The reviewers appreciated the paper's ideas; however, all the reviewers were on the fence with borderline scores. In summary, this is a borderline paper, and unfortunately, the final decision is a rejection. The reviewers have provided detailed and constructive feedback for improving the paper. In particular, the authors should incorporate the reviewers' feedback to better position the work w.r.t. the existing literature and provide clear reasoning behind the gains for KD+ in experiments. This is exciting and potentially impactful work, and we encourage the authors to incorporate the reviewers' feedback when preparing future revisions of the paper.